Force-velocity relationship profile of elbow flexors in male gymnasts

Nakatani Miyuki 1 2
Murata Kensuke 2
Kanehisa Hiroaki 3
Takai Yohei y-takai@nifs-k.ac.jp 2
1 The Center for Liberal Arts, Meiji Gakuin University , Yokohama , Kanagawa , Japan
2 National Institute of Fitness and Sports in Kanoya , Kanoya , Kagosima , Japan
3 Faculty of Sport and Health Science, Ritsumeikan University , Kusatsu , Shiga , Japan
García-Ramos Amador
Electronic publication date: 2021 Mar 15
Publication date: 2021
Volume: 9
Electronic Location ID: e10907
Received 2020 Sep 23; Accepted 2021 Jan 14
Copyright: ©2021 Nakatani et al.
Copyright year: 2021
Copyright holder: Nakatani et al.
License: This is an open access article distributed under the terms of the Creative Commons Attribution License, which permits unrestricted use, distribution, reproduction and adaptation in any medium and for any purpose provided that it is properly attributed. For attribution, the original author(s), title, publication source (PeerJ) and either DOI or URL of the article must be cited.
License URL: https://creativecommons.org/licenses/by/4.0/

Keywords: Power, Muscle thickness, Maximal voluntary contraction, EMG, Sport specific, Judo athlete, Biceps brachii, Dynamic task

Funding: The authors received no funding for this work.

==============================
Background

The theoretical maximum force (F0), velocity (V0), and power (Pmax) of athletes calculated from the relationship between force and velocity (F-V relationship) and the slope of the F-V relationship, reflect their competitive and training activity profiles. Evaluating the F-V relationship of athletes facilitates categorizing the profiles of dynamic muscle functions in relation to long-term sport-specific training. For gymnastics, however, no studies have tried to examine the profiles of F-V relation and power output for upper limb muscles in relation to the muscularity, while the use of the upper extremities in this sport is very unique as described earlier.

Purpose

It was hypothesized that the F-V relationship of the elbow flexion in gymnasts might be characterized by low capacity for generating explosive force, notably in terms of the force normalized to muscle size.

Methods

The F0, V0, and Pmax derived from the force-velocity relationship during explosive elbow flexion against six different loads (unloaded condition, 15, 30, 45, 60, and 75% of maximal voluntary isometric elbow flexion force (MVFEF)) for 16 gymnasts (GYM) and 22 judo athletes (JD). F0 and Pmax were expressed as values relative to the cross-sectional area index (CSAindex) of elbow flexors (F0/CSAindex and Pmax/CSAindex, respectively), which was calculated from muscle thickness in the anterior upper arm. The electromyogram (EMG) activities of the biceps brachii (BB) during the maximal isometric and dynamic tasks were also determined.

Results

There were no significant differences in CSAindex of elbow flexors between GYM and JD. MVFEF/CSAindex for GYM was significantly lower than that for JD. Force was linearly associated with velocity in the dynamic elbow flexion for all the participants (r =  − 0.997 to −0.905 for GYM, r =  − 0.998 to −0.840 for JD). F0, F0/ CSAindex, V0, Pmax, Pmax/CSAindex, and MVFEF were significantly lower in GYM than in JD. The activity levels of BB during the dynamic tasks tended to be lower in GYM than in JD at load of <45%MVC.

Conclusion

Gymnasts cannot generate explosive elbow flexion force corresponding to their muscle size. This may be due to low neuromuscular activities during the maximal dynamic tasks against relatively low loads.

Introduction

The competitive events of artistic gymnastics for men consist of “floor,” “rings,” “pommel horse,” “long horse,” “parallel bars,” and the “horizontal bar.” Gymnastic training involves, on average, 102 impacts per session, and loads of 1.5 to 3.6 times the bodyweight on the upper extremity when performing the actions such as hurdle step, round-off, back handspring, forward handspring, and pommel of young gymnasts (Daly et al., 1999). During the handstand and the swallow on the rings, the electromyogram amplitude of the biceps brachii, normalized to that during maximal voluntary contraction (MVC) is as high as 50–80% (Bernasconi et al., 2009; Kochanowicz et al., 2018b). Gymnasts are frequently required to support their body mass and control body balance by using the upper extremities while overcoming repetitive high-impact loadings (DiFiori et al., 2002). In other words, gymnasts repeat highly intense and sustained upper arm muscle activities during competitions and training. The unique use of upper limb muscles by gymnasts is one factor yielding the hypertrophied muscularity of this segment (Claessens et al., 1991; Ichinose et al., 1998; Spenst, Martin & Drinkwater, 1993; Takai et al., 2018)

The muscle size (e.g., muscle cross-sectional area and muscle volume) is a significant determinant of force- and power-generating capacities of the upper arms (Fukunaga et al., 2001; Wakahara et al., 2013). There is little information from earlier studies on the isometric and dynamic strength of the upper limb muscles of gymnasts. Only three studies have provided data on isometric and dynamic strength of gymnasts (Kochanowicz et al., 2018b; Kochanowicz et al., 2019; Niespodzinski et al., 2018), but their findings are mutually contradictory. One study has found higher isometric elbow flexor strength in male gymnasts compared to untrained people (Niespodzinski et al., 2018), but other studies have reported the opposite result (Kochanowicz et al., 2018b; Kochanowicz et al., 2019). The earlier studies have attempted to clarify force-generating capacity of gymnasts compared to individuals who have not experienced regular sport-specific training. In general, well-trained individuals have greater muscle size as well as voluntary strength compared to sedentary individuals (Alway et al., 1990; Sale et al., 1987). For clarifying the profiles of force- and power-generating capacities in gymnasts, therefore, it is necessary to compare them with well-trained individuals with similar upper limb muscularity as that of gymnasts.

Many studies aiming to evaluate the dynamic muscle function of athletes have determined the force-velocity (F-V) and/or the load-power relationship of explosive multi-joint movements such as the bench press, throwing, jumping, and cycling, which is obtained by using loads relative to one repetition maximum (1RM) of the task or body mass (Asci & Acikada, 2007; Baker, 2001; Baker & Newton, 2006; Bozic & Bacvarevic, 2018; Giroux et al., 2016; Izquierdo et al., 2002; McBride et al., 1999; Vuk, Markovic & Jaric, 2012). Their findings suggest that the theoretical maximum force (F0), velocity (V0), and power (Pmax) of athletes calculated from the F-V relationship and the slope of the F-V relationship, reflect their competitive and training activity profiles (Bozic & Bacvarevic, 2018; Giroux et al., 2016; Izquierdo et al., 2002; McBride et al., 1999). For example, Bozic & Bacvarevic (2018) found that in maximal sprints on a leg cycle ergometer, wrestlers and judo athletes showed higher F0 with force-oriented slope, which means steeper slope, and the sprinters higher V0. Evaluating the F-V relationship of athletes facilitates categorizing the profiles of dynamic muscle functions in relation to long-term sport-specific training. For gymnastics, however, no studies have tried to examine the profiles of F-V relation and power output for upper limb muscles in relation to the muscularity, while the use of the upper extremities in this sport is very unique as described earlier.

Ballistic and/or explosive exercises are highly useful for improving power production (Cormie, McGuigan & Newton, 2011). However, such training-induced changes in maximal power production and F-V relationships vary with the magnitude of the adapted load and the actual movement velocity during exercise (Cormie, McGuigan & Newton, 2011; Djuric et al., 2016; Jimenez-Reyes et al., 2016; Kaneko et al., 1984; McBride et al., 1999). As described above, competitive and training activities for gymnasts can be characterized as highly intense and sustained muscle contractions to support the body mass and the successful control of body balance. A training modality with intense and sustained muscle contractions (lasting 3 s at 75% of MVC) is less effective for explosive muscle functions and activation compared to explosive contractions at >80% of MVC lasting <1 s (Balshaw et al., 2016). No significant difference in isometric MVC torque of elbow flexion has been reported between gymnasts and untrained individuals, in spite of greater arm lean tissue mass in gymnasts (Kochanowicz et al., 2018b). Based on these findings, we can hypothesize that as a result of long-term sport-specific training, the F-V relationship of the upper limb muscles in gymnasts might be characterized by low capacity for generating explosive force, notably in terms of the force normalized to muscle size, i.e., muscle quality. This study aimed to clarify the profile of the F-V relationship of elbow flexors in male gymnasts.

Methods

Participants

Thirty-eight adult men voluntarily participated in this study. The means and standard deviations (SDs) for age, body height, and body mass were 20.7 ± 1.2 years, 167.0 ± 5.2 cm, and 68.8 ± 7.5 kg, respectively. As shown in Table 1, the participants were divided into two groups: gymnasts (GYM; N = 16) and judo athletes (JD; N = 22). Judo athletes as well as gymnasts are characterized by a predominant muscular development in the upper limb (Claessens et al., 1991; Ichinose et al., 1998; Spenst, Martin & Drinkwater, 1993; Takai et al., 2018). Thus, we adopted judo athletes as a control group. GYM was significantly shorter and lighter than JD. All participants had experienced competitive activities and systematized physical training programs in their major sport for eight or more years. They had competed in intercollegiate or international athletic meetings in the preceding year. The ethical committee of the local university approved this study (the National Institute of Fitness and Sports in Kanoya’s Ethics Committee #11-102). We conducted the study consistent with the requirements for human experimentation in the Declaration of Helsinki. We informed all participants about the purpose and procedures of this study and possible measurements risks before the experiment. All the participants gave their written informed consent for participation in the study.

Table 1 Physical characteristics of the participants.

Variables	GYM, n = 16	JD, n = 22	p	Cohen’sd	
Height, cm	163.0	±	4.0	170.9	±	6.5	<0.001	1.47	
Body mass, kg	58.9	±	2.8	78.8	±	12.2	<0.001	2.24	
Upper arm length, cm	30.6	±	1.3	32.4	±	1.8	0.001	1.17	
MTant, cm	3.6	±	0.3	3.5	±	0.4	0.212	0.43	
MTpos, cm	4.3	±	0.5	4.8	±	0.4	0.005	0.98	
CSAindex of elbow flexor	10.4	±	1.6	9.6	±	2.2	0.255	0.39	
CSAindex of elbow extensor	15.0	±	2.9	18.1	±	3.4	0.005	0.99	
MVFEF, N	242.5	±	23.6	284.8	±	45.8	0.001	1.16	
MVFEE, N	201.0	±	47.8	262.9	±	79.2	0.005	0.95	
MVFEF/CSAindex, N/cm2	23.7	±	3.0	30.4	±	5.3	<0.001	1.52	
MVFEE/CSAindex, N/cm2	13.7	±	2.9	14.8	±	4.9	0.411	0.28	
Notes.

Values are means ± SDs.

MTant, muscle thickness at upper arm anterior.

MTpos, muscle thickness at upper arm posterior.

CSAindex, muscle cross-sectional area index obtained using the equation of π × (MT∕2)2.

MVFEF, maximal voluntary isometric elbow flexion force.

MVFEE, maximal voluntary isometric elbow extension force.

Experimental design

In addition to the anthropometric and ultrasound measurements, all participants were involved in maximal voluntary isometric and dynamic contraction tasks. Firstly, anthropometry and ultrasound measurements were conducted. After the standardized warm-up and familiarization with measurement apparatus, the participants were encouraged to perform maximal voluntary isometric contraction (MVC) task, followed by dynamic contraction task, in elbow flexion. After a 5-min rest following the completion the isometric MVC tasks, the dynamic contraction task was conducted. During the tasks, the electromyogram (EMG) activities of elbow flexors and extensors were recorded. All measurements were conducted by the same investigator (MN).

An earlier finding has demonstrated that the elbow flexion strength is greater in gymnasts than in untrained individuals, but not in elbow extension strength (Niespodzinski et al., 2018). This suggests that gymnastic training would improve the strength capability of the elbow flexors more than that of elbow extensors. Therefore, we examined the F-V relation the elbow flexors in gymnasts.

Measurements of muscle thickness (MT)

We measured the MTs in the anterior (MTant) and the posterior (MTpos) part of the upper arm as variables representing the size of elbow flexors and extensors, by using a brightness-mode ultrasound apparatus (ProSound Alpha6, Hitachi Aloka Medical, Japan) with a linear-array probe (7.27 MHz). The procedure for obtaining ultrasonographic images and for determining MT from the images was identical to that described in an earlier study (Abe et al., 1994). Briefly, the MT measurements for the two sites were conducted at 60% of the upper arm length defined as the distance from the acromial process to the lateral epicondyle of the humerus. During the measurements, the subjects stood upright with their arms relaxed and extended. The probe was placed perpendicular to the skin without depressing the dermal surface and a probe was coated with water-soluble transmission gel, which provided acoustic contact. The MT was defined as the distance from the subcutaneous adipose tissue-muscle interface to the muscle-bone interface. The upper arm anterior and posterior MTs were referred to as MTant and MTpos, respectively. The muscles involved in the MTant were the biceps brachii and brachioradialis and that in the MTpos was the triceps brachii. All images were analyzed by using image analysis software (Image J ver. 1.47, NIH, USA). We calculated muscle cross-sectional area index (CSAindex) of the elbow flexors and extensors by using the following equation (Miyatani, Kanehisa & Fukunaga, 2000):

CSAindex = π × (MT/2)2

where π is a constant, 3.14159, and MT is MTant or MTpos in cm. The reproducibility of the MT measurements was assessed on 2 separate days (with an interval of >4 d) in a pilot study with 7 young adults (25.0 ± 2.6 yr, 166.7 ± 8.7 cm, and 65.0 ± 7.6 kg). For MTant and MTpos, there were no significant differences in the mean values between the first and second measurement. The reproducibility of the MT measurements in this study were 1.5–4.1% for CV and 0.911 to 0.976 for ICC.

Experimental setup for maximal isometric (MVC) and dynamic contraction tasks

All the participants performed the MVC and the dynamic contraction elbow flexion tasks with the right arm using a custom-made dynamometer with tension/compression load cells (TR22S, SOHGOH KEISO CO., LTD, Japan) as shown in Fig. 1. Participants were seated on an adjustable chair with the shoulder, and hip joints flexed at 90°. Their hips and shoulders were fixed to backrests of chairs, and wrists were fixed to lever arms of the dynamometer in a neutral position by non-elastic belts. The rotation axis of the elbow joint was visually aligned as closely as possible with that of the dynamometer. The forearm was fixed to the lever arm that could rotate freely around the axis with the wrist joint kept in a neutral position. The force signals during the tasks were amplified and attenuated with a low-pass filter (<100 Hz, DPM-912B, KYOWA, Japan). The axis of the potentiometer’s lever arm was equipped with a dynamometer to detect voltage changes associated with those in the elbow joint angles during the dynamic contraction task. The voltage signals were converted to angle (deg) from the voltage-angle relationship. The force and angle signals were sampled at a frequency of 2 kHz via a 16-bit analog/digital converter (PowerLab/16s: AD Instruments Sydney, Australia) and stored on a personal computer.

Figure 1 Experimental setup for maximal isometric (MVC) and dynamic contraction tasks.

Schematic diagram of the experimental set up for conducting the maximal isometric (MVC) and dynamic contraction tasks. The participants sat on a chair adjusted for the testing position. Their right arms were fixed to the dynamometer with the shoulder flexed at 90° and the forearm in a neutral position.

MVC task

Submaximal contractions were conducted as a warm-up exercise. Then, before the dynamic contraction task, the participants conducted the MVC tasks by flexing and extending each elbow joint by gradually exerting elbow flexion or extension force from the baseline to the maximum level, and sustained it at the maximum for approximately 2 s. The elbow joint was held at a 40° flexed position (0° corresponds to full elbow extension). After a standardized warm-up protocol (50% and 80% of subjective effect) and familiarization with the measurement apparatus, two trials were performed with a 3-min interval between trials. If the difference between the isometric forces of the two trials was more than 10%, the measurement was made again. The highest value among the 2 or 3 isometric forces was adapted as the elbow flexion (MVFEF) or extension (MVFEE) MVC force. The MVFEF was used to determine the load set in the dynamic contraction task.

Dynamic contraction task

After a 5-min rest following the completion the MVC tasks, the participants were asked to perform the dynamic contraction task consisting of ballistic contractions against six different loads in a random order (unload condition and 15, 30, 45, 60, 75% of MVC). They were asked to flex the elbow joint as strongly and quickly as possible in each of the six load conditions. The participants’ position and the fixation of the body during the dynamic contraction task were identical to those during the MVC tasks. Weights were attached to pulley moving in conjunction with the lever arm, and the range of the motion was from 40° to 120° of the elbow joint angle. A shock absorber was put on the portion at 120°. Before each trial, and an examiner lifted the lever arm until the start position (corresponded to 40°) on checking raw data of joint angle with a monitor visually. At the starting position, the participants were kept to relaxed condition by supporting the load by the examiner until the start of elbow flexion with maximal effort. Participants were informed that the magnitude of the load had been set in advance. Rest intervals of 1 min and 3 min respectively were set between trials in a given load condition and between loads sets. The analysis of elbow flexion force and velocity at each load condition is described in detail bellows.

Recordings of electromyograms (EMGs)

Surface EMGs were recorded during the MVC and dynamic contraction tasks from the brachioradialis (Bra), the short head of biceps brachii (BB), and the long head of the triceps brachii (TB) by using bipolar Ag-AgCl electrodes (F-150S, NIHON KOHDEN Corp., Tokyo, Japan) along the direction of the muscle fascicles. Bipolar electrodes (5 mm diameter, 20 mm interelectrode distance) were placed over the muscle bellies after the skin surface was shaved and rubbed with sandpaper and cleaned with alcohol. The electrodes were connected to a differential amplifier (×1000) with a bandwidth of 5-1000 Hz. (MEG-6100, NIHON KOHDEN Corp., Tokyo, Japan) The EMG signals, as well as force and angle signals, were stored on a personal computer via an analog-to-digital converter (PowerLab/16s: AD Instruments Sydney, Australia) at a sampling rate of 2 kHz. The trial in which the highest MVC force appeared was adopted to analyze the EMG data of every muscle in the MVC task.

We attenuated the EMG amplitude by using a first-order Butterworth high-pass filter (>300 Hz) with a zero-phase lag before rectification, which was following by a first-order Butterworth low-pass filter at 5 Hz with a zero-phase lag (Yoshitake et al., 2014). We rectified the EMG amplitude during the MVC task and averaged the amplitude over a 1-s window centered at the time when the peak force appeared, which was normalized to this value during the dynamic contraction task. The analysis of the EMG amplitude during the dynamic contraction task is described in detail below.

Velocity, power, and EMG amplitude during dynamic contraction

Figure 2 shows typical examples of dynamic contraction tasks when unloading, at 30% and 75% MVFEF in one gymnast. We obtained the angular velocity by differentiating the angle by time. Then, we converted it to the tangential velocity (the elbow flexion velocity, m/s) by multiplying the perpendicular distance between the load cell and the lever-arm axis of the dynamometer. We calculated the power by multiplying the exerted force by the velocity. We averaged each variable over a range of elbow joint angles from 40° to 100° and used as functional variables developed for the specific load condition. We referred to the force and velocity as F and V, respectively, and we obtained the mean power (P) from the product of F and V. In addition to the absolute values, we expressed F and P as values relative to CSAindex (F/CSAindex and P/CSAindex, respectively). The mean values of the filtered EMG for each of the three muscles were expressed as the value relative to the EMG amplitude during the MVC task (%EMGMV C).

Figure 2 Typical examples of dynamic contraction tasks.

Typical examples of the elbow joint angle (A), force (B), velocity (C), power (D), and the EMG amplitude of BB (E) during the dynamic contraction task when unloading, 30% and 75% MVFEF for one gymnast.

Calculation of the theoretical maximal force (F0), velocity (V0), and power (Pmax)

We calculated the F0, V0, and Pmax as basic indicators of the relationship between F and V (F-V relationship) across the six different loads (Fig. 3). We defined the points of intersection of the regression line with the ordinate and transversal axis as F0, and V0, respectively, and calculated Pmax as described in an earlier study (Jaric, 2015; Samozino et al., 2012; Vandewalle et al., 1987) by using the following equation:

Figure 3 Force-velocity relationship and parameters.

The average values (A) and individual values (B). Force-velocity relationship and parameters derived from each of the two relationships of gymnasts (the closed circle) and judo athletes (the open circle).

Pmax = F0 × V0 / 4.

In addition to the absolute values, we expressed F0 and Pmax as values relative to CSAindex (F0/CSAindex and Pmax/CSAindex). Furthermore, we adopted the slope of the regression line for the F-V relationship (F-Vslope) as a parameter indicative of predominance of force (or velocity) in the relationship (Samozino et al., 2012). To evaluate the test-retest reliability of ballistic power testing, each subject was tested on 2 separate occasions at the same time of day after an interval at least 3 days. The same warm-up routine and testing protocol were used in both occasions. To determine the test-retest reliability across the two testing sessions, the intraclass correlation coefficient (ICC 1,1) was used. There was no significant difference between the two testing sessions in each of F0, V0 and Pmax. The ICC(1,1) for each of the measured parameters ranged from 0.820 to 0.984.

Statistics

We have presented descriptive data as means ± SDs. We used an unpaired Student’s t-test to examine differences in measured variables between GYM and JD, and a two-way repeated measures analysis of variance (ANOVA: 2 groups ×6 loads) to test the main effects of group and load and their interaction on %EMGMV C for the examined muscles. When appropriate, we used simple main effect test was used to test the significance of the group difference for post hoc comparison. We calculated Pearson’s product-moment correlation coefficient (r) to examine the associations between F and V. We also calculated Cohen’s d (for a post hoc test) and η2 (for ANOVA) as indices of effect sizes. We interpreted Cohen’s d as large: ≥0.80, medium: 0.50–0.79, small: 0.20–0.49, or trivial: <0.20, and we interpreted η2 was as large: 0.14, medium: 0.06, or small: 0.01 (Cohen, 1988). Sphericity was checked by Mauchly’s test in ANOVA, and p values were modified with Greenhouse–Geisser correction when necessary. We set the level of significance as p < 0.05. We analyzed all the data using SPSS software (SPSS statistics 25; IBM, Japan).

Results

There were no significant differences in MTant and CSAindex of elbow flexor between GYM and JD, although MTpos and CSAindex of elbow extensor were significantly smaller in GYM than in JD (Table 1). MVFEF/CSAindex for GYM was significantly lower than that for JD, while the corresponding difference was not found in MVFEE/CSAindex.

Figure 3 shows an example of F-V relationship. F was linearly associated with V in all the participants (r =  − 0.997 to −0.905 for GYM, r =  − 0.998 to −0.840 for JD). Each of the theoretical maximum parameters was significantly lower in GYM than in JD (Table 2). In addition, the F-Vslope was steeper in GYM compared to JD. The F0/CSAindex and Pmax/CSAindex were significantly lower in GYM than in JD (Table 2).

Table 2 Descriptive data on the parameters derived from force-velocity relation of elbow flexors.

Variables	GYM, n = 16	JD, n = 22	p	Cohen’sd	
F0, N	260.9	±	47.1	311.5	±	63.0	0.010	0.89	
V0, m/s	1.5	±	0.4	2.2	±	0.3	<0.001	2.11	
Pmax, W	96.3	±	23.9	173.2	±	41.6	<0.001	2.17	
F-Vslope	−190.5	±	91.2	−143.3	±	39.1	0.036	0.72	
F0/CSAindex, N/cm2	25.3	±	3.6	33.0	±	5.8	<0.001	1.54	
Pmax/CSAindex, W/cm2	9.4	±	2.4	18.3	±	3.9	<0.001	2.63	
Notes.

Values are means ±SDs.

F0, theoretical maximal force.

V0, theoretical maximal velocity.

Pmax, theoretical maximal power.

F-Vslope, slope of the regression line for the relationship between force and velocity.

CSAindex, muscle cross-sectional area index obtained using the equation of π × (MT∕2)2.

A two-way ANOVA indicated neither a significant interaction between %EMGMV C and load nor a significant main effect of group for Bra (p = 0.173, η2 = 0.206) and TB (p = 0.563, η2 = 0.481): 125.9 ± 49.2% for Bra, and 6.8 ± 2.5% for TB in GYM and 120.1 ± 32.6% for Bra, and 9.7 ± 8.3% for TB in JD. For BB, however, the ANOVA revealed a significant interaction (p = 0.017, η2 = 0.080). The %EMGMV C of BB at unload condition was lower in GYM than in JD (p = 0.022, Cohen’s d = 1.41). In addition, the %EMGMV C values of BB at 30 and 40%MVC conditions tended to be lower in GYM compared to JD (p = 0.069-0.083, Cohen’s d = 0.663-0.923).

F0, V0, Pmax and F-Vslope were significantly lower in GYM (260.9 ±47.1 N, 1.5 ± 0.4 m/s, 96.3 ± 23.9 W, −190.5 ± 91.2) than in JD (311.5 ± 63.0 N, 2.2 ± 0.3 m/s, 173.2 ± 41.6 W, -143.3 ± 39.1).

Discussion

The main findings obtained here were that (1) GYM had lower F0, V0, Pmax, and F-Vslope than JD, (2) GYM had lower MVFEF/CSAindex and F0/CSAindex than JD, and (3) the activity levels of BB during the dynamic tasks tended to be lower in GYM than in JD at load of <45%MVC. The regression line slope of the F-V relationship in athletes reflects their competitive and training activity profiles, and it becomes a parameter for discriminating force- or velocity-oriented type of athletes (Bozic & Bacvarevic, 2018; Giroux et al., 2016; Izquierdo et al., 2002; McBride et al., 1999). Thus, the result on F-Vslope indicates that as compared to JD, gymnasts show a force-orientated profile in explosive elbow flexion. Furthermore, the second result supports the hypothesis that the F-V relationship of elbow flexors in gymnasts is characterized by the low capacity for generating an explosive force relative to muscle size. In addition. The third result implies that the observed force-orientated profile and low V0, F0/CSAindex, and Pmax in GYM might be partially attributable to low activation of elbow flexors during explosive dynamic contractions in this population, notably in conditions requiring quick contraction against light loads.

There are three possible explanations for the force-oriented profile and the lower power generating capacity in GYM compared to JD. (1) An imbalance between morphological adaptation and neural adaptation of the elbow flexors caused by long-term gymnastic training; (2) lower muscular activation during explosive elbow flexion; and (3) increased hypertrophied muscles relative to limb length. Firstly, as described earlier, the activities of upper limb muscles during gymnastics can be characterized by highly intense and sustained contractions and/or co-contractions between the agonist and antagonistic muscles. Prolonged maximum voluntarily co-contraction training produces a significant gain in muscle size without an improvement in muscle strength (Maeo et al., 2014). Mitchell et al. (2012) have proposed that training-induced gains in the muscle volume of the quadriceps femoris were similar between training programs with 30% and 80% of 1RM to failure, but isotonic maximal strength gain was more significant in high-intensity than in low-intensity programs. These findings suggest that a training modality with long-term sustained contractions would result in an imbalance between hypertrophic and neuromuscular adaptations of exercising muscles. Furthermore, Kochanowicz et al. (2018b) reported no significant difference in elbow flexion strength between gymnasts and untrained individuals, whereas gymnasts had a greater lean tissue mass in the arms than untrained individuals. Cross-sectional studies have also provided evidence that dynamic strength normalized to the muscle size of body-builders, who are generally categorized as the practitioners of high-volume resistance exercises (Hackett, Johnson & Chow, 2013), is lower at the whole muscle (Alway et al., 1990; Sale et al., 1987) and single muscle fiber (Meijer et al., 2015) levels than in non-athletes or power athletes. Taken together, it is likely that long-term participation in gymnastics training produces a relatively higher muscle size gain than isometric or dynamic strength, and consequently causes the low F0/CSAindex in gymnasts, i.e., muscle quality.

Secondly, the muscular activities of BB during explosive elbow flexion at relatively low load tended to be lower in GYM than in JD, whereas no significant group difference in submaximal EMG amplitude during isometric contraction was found in this study (Supplemental data). Combined this with the current finding, the lower muscular activities during dynamic contraction task in GYM may be explained as a result of sport-specific adaptation in the BB of this athletic group. Agonist muscle activation in the early phase of explosive torque development is strongly associated with the initial torque output in isometric knee extension contractions (De Ruiter et al., 2004; De Ruiter et al., 2006; De Ruiter et al., 2007). Highly intense and sustained training elicits muscle hypertrophy (Massey et al., 2018) and attenuates the activation level in the earlier phases of force development during explosive isometric knee extensions (Balshaw et al., 2016; Tillin & Folland, 2014). Furthermore, training modalities with slow movements and tonic force generation that causes sustained muscular activity increases isometric strength and muscle size (Tanimoto & Ishii, 2006), but has little effect on dynamic strength and power production (Tanimoto & Ishii, 2006; Usui et al., 2016). Considering these findings, lower muscular activation level of BB during explosive elbow flexion in gymnasts might be due to type of training modality in gymnasts.

Thirdly, GYM had higher ratios of CSAindex and MTant to upper arm length: 0.34 ± 0.06 cm2/cm for GYM vs. 0.30 ± 0.06 cm2/cm for JD in CSAEF (p = 0.032, Cohen’s d = 0.73) and 0.12 ± 0.01 cm/cm for GYM vs. 0.11 ± 0.01 cm/cm for JD in MTant (p = 0.003, Cohen’s d = 1.05). The mean values of the ratio of MTant to upper arm length in GYM and JD were higher by 18% and 5%, respectively, compared to reference data obtained from the general Japanese population (Wakahara et al., 2010), which indicates that GYM has a larger elbow flexor muscle size for a given upper arm length. Most fibers of elbow flexors have equal length and uniform thickness (Kaufman, An & Chao, 1989). The fibers in this muscle group are attached to a tendon plate that extends into the muscle belly and organizes a large number of fibers with similar length and thickness in parallel, which is called the “parallelepipedon” (An et al., 1981). When a muscle is hypertrophied, the length of the tendon plate appears to be extended further into the muscle belly, and the fibers must pull at a more oblique angle to the direction of induced motion (the line of pull of the tendon end) (Kaufman, An & Chao, 1989). Therefore, the fiber alignment is more oblique to the force loss in the line of action. The influence of this could be greater at higher contraction velocities (Maughan, Watson & Weir, 1984). Therefore, the low F0/CSAindex in GYM might be caused by the morphological profile of elbow flexor muscles that is characterized by a high ratio of muscle size to upper limb length.

In addition to the aforementioned aspects, the influence of fiber composition might also be involved to explain why GYM showed lower F0, V0 and Pmax than JD. It is known that a 14-week resistance training of the quadriceps femoris yields a reduction in the relative portion of type IIX muscle fiber, and its decline negatively influenced the rate of force development in the early phase (<100 ms) (Andersen et al., 2010). Furthermore, Kesidis et al. (2008) observed lower percentage of type IIX fiber for the vastus lateralis in bodybuilders than in physical education students. If these findings can be applied to the current results, there is a possibility that low V0 in GYM compared to JD might be due to the group difference in the percentage of type IIX fiber.

There are some limitations in this study. Firstly, we determined MT as a measure of muscle size and used CSAindex calculated from MT to normalize F. Miyatani, Kanehisa & Fukunaga (2000) reported that the sum of the product of CSAindex and upper arm length for the elbow flexors and extensors strongly correlated with the MRI-based muscle volumes of the two muscle groups (r = 0.962). These findings indicate that either MT or CSAindex adopted here can be qualitative parameters of a specific muscle group, although the previous studies have not examined the direct associations of these variables with the muscle CSAindex of the elbow flexors. At the same time, the reports of Miyatani, Kanehisa & Fukunaga (2000) warrants to interpret the current results as that the muscle quality of elbow flexors in GYM is lower than that in JD. Secondly, the muscle activities during handstand are higher in the elbow extensors than in the elbow flexors (Kochanowicz et al., 2018a). Furthermore, F-V profile may be affected by muscle architecture (Morales-Artacho et al., 2018). The elbow flexors are mainly consisted of parallel muscles and the elbow extensors are pennate muscles. Therefore, the F-V profile of the elbow extensors would be different from that of the elbow flexors. Thirdly, it is known that force-velocity profile of the upper body differs between men and women (Torrejón et al., 2019). We have no data concerning the force-velocity profile of female gymnasts. Hence, we cannot conclude whether the current findings are applied to female gymnasts. Further investigations are needed to clarify these points.

Practical application

The current findings indicate that gymnasts cannot generate explosive elbow flexion force corresponding to their muscle size. This may be due to low neuromuscular activities during the maximal dynamic tasks against relatively low loads. As described earlier, gymnasts are frequently required to support their body mass and control body balance by using the upper extremities while overcoming repetitive high-impact loadings (DiFiori et al., 2002). This implies that regardless of elbow flexors and extensors, to gain the explosive force generation capability of the upper limb muscles will be a factor for improving gymnastic performance. Training-induced changes in muscle functions and activation in the early phase of force development depend on the type of muscle contraction (sustained vs. explosive) (Balshaw et al., 2016; Massey et al., 2018; Tillin & Folland, 2014), load adapted, and contraction velocities (Kaneko et al., 1984). Ballistic and/or explosive exercises can greatly improve power production (Cormie, McGuigan & Newton, 2011). On the other hand, a training modality with intense and sustained muscle contractions is less effective for explosive muscle functions and activation compared to that consisting of explosive exercise (Balshaw et al., 2016). Taking these aspects into account together with the findings obtained here, it will be recommended for gymnasts and their coaches that for improving explosive force generation capacity of the elbow flexors, training program including ballistic and/or explosive exercises for this muscle group should be involved to the schedule of their regular training activities.

Conclusions

The current findings demonstrate that as compared to judo athletes, gymnasts have a force-oriented profile and low capacity for generating explosive force in elbow flexors, which is partially due to neuromuscular activity during explosive elbow flexion against relatively low load and force exerted normalized to muscle size.

Supplemental Information

Supplemental Information 1 Force-velocity profile in male gymnasts

Click here for additional data file.

The authors wish to express their gratitude to the students of the National Institute of Fitness and Sports in Kanoya for their contribution to this study. This study was supported by a NIFS project for the assessment of physical fitness for athletes.

Additional Information and Declarations

Competing Interests

Author Contributions

Human Ethics

Data Availability

The authors declare there are no competing interests.

Miyuki Nakatani and Yohei Takai conceived and designed the experiments, performed the experiments, analyzed the data, prepared figures and/or tables, authored or reviewed drafts of the paper, and approved the final draft.

Kensuke Murata performed the experiments, authored or reviewed drafts of the paper, and approved the final draft.

Hiroaki Kanehisa conceived and designed the experiments, analyzed the data, prepared figures and/or tables, authored or reviewed drafts of the paper, and approved the final draft.

The following information was supplied relating to ethical approvals (i.e., approving body and any reference numbers):

The National Institute of Fitness and Sports in Kanoya’s Ethics Committee approved this research (#11-102).

The following information was supplied regarding data availability:

Raw data is available as a Supplemental File.

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
