# Peer review of "Force-velocity relationship profile of elbow flexors in male gymnasts"

_PeerJ, doi:10.7717/peerj.10907_

## Round 0.1 · original submission · Major Revisions

Reviewers provided generally positive comments, but some changes are required.

Reviewer 1 ·

Basic reporting

The academic language was very good and abstract was very well constructed.

However, it would be nice to check the space after/before the parenthesis in the introduction part as well as the underlined word Line 75. What was the idea to compare between Judo and gymnasts?

Table 1 and 2 should it be p< 0.001 instead of 0.000?

What did mean for "b, p-value " in the Table 1?

Why the number of the participants in each group was not similar for Table 1 and 2?

Please identify the tested movement on Table 2 ....F-V elbow flexion?

Did the value in Figure 3A being an average for all which should be in the Fig3B?

Experimental design

The study focused the structural and muscle function adaptation on gymnasts which the results gains in on these questions and methods was indeed easy to follow and all measures were provided the reliability.

However. there are still unclear some points.

Was there the same investigator in all measurement?

Line 138 shouldn't be BB and brachialis muscle at this MT measurement point as the same as (Miyatani et al. 2000)? and from that study, why this study did concern using CSA to normalize F instead of MV?

Why the MVC task was set at 40 deg for both elbow flexion and extension? Shouldn't be at the optimal angle?

Why the dynamic contraction task was done only for EF?

Validity of the findings

Result, it would be interesting to present the value of EMGMVC or submax perhaps in the Table1.

In the statistic part, the pearson r correlation was used, but why in the result the R2 was presented. Please clarify.

The report of F0, V0, Pmax, F-V slope should be partly presented in the result part.

The second last paragraph of discussion, Is there any document for EF in the relation of muscle fibre type and RFD?

Perspective, please check the "and" after the citations.

Additional comments

-

Annotated reviews are not available for download in order to protect the identity of reviewers who chose to remain anonymous.

Reviewer 2 ·

Basic reporting

No comment.

Experimental design

No comment.

Validity of the findings

No comment.

Additional comments

- The introduction while interesting should be clear and concise and be hypothesis driven to allow the reader to see the basis of your hypothesis. At present this could be improved. For example, in lines 70-76, authors must indicate what can explain these contradictory results.

- The authors need to clarify the purpose of the study.

- The methods need to be pristine so that the study can be replicated.

- A session describing the experimental design can help readers. In addition, this session may present the interval between the evaluations and the order that they were carried out.

- Why was the study conducted only with men?

- Why were reproducibility measures performed only for muscle thickness?

- The practical applications are for the coach or end user and needs to be clearer and more to the point.

- The discussion needs to reflect what you found, how it relates to the literature and then what it means physiologically.

- Authors should avoid speculation at the conclusion of the study.

---

## Round 0.2 · accepted · Accept

Congratulations for meeting the high standard of PeerJ

Reviewer 2 ·

Basic reporting

No comments.

Experimental design

No comments.

Validity of the findings

No comments.

Additional comments

The authors responded to all my comments accordingly and considerably improved the manuscript.